# Progress of Polymer Application in Coated Proppant and Ultra-Low Density Proppant

**DOI:** 10.3390/polym14245534

**Published:** 2022-12-17

**Authors:** Tao Chen, Jie Gao, Yuan Zhao, Tian Liang, Guowen Hu, Xiaobing Han

**Affiliations:** Hubei Key Laboratory of Radiation Chemistry and Functional Materials, School of Nuclear Technology and Chemistry & Biology, Hubei University of Science and Technology, Xianning 437100, China

**Keywords:** LDP, polymer, sand, ceramics, nutshell, ULDP, fabrication, property

## Abstract

Design, synthesis and application of low-density proppant (LDP) are of great significance for efficient and clean exploitation of low permeability oil and gas. On the basis of a brief introduction of hydraulic fracturing and the application of traditional proppants, this review systematically summarized the polymer application progress in LDP, including coated sand, coated ceramics, coated nutshells, especially for polymer composites based ultra-low density proppant (ULDP). Finally, the existing problems and future development direction are also prospected.

## 1. Introduction

As one of the most important technology for the exploitation of oil and gas, hydraulic fracturing has been widely used in the exploitation of unconventional low permeability reservoir, and secondary exploitation of old oil and gas wells (Figure 1) [1]. Hydraulic fracturing can stimulate hydrocarbon production by creating a network of highly conductivity fractures surrounding a wellbore. In the hydraulic fracturing, proppant is brought into the fractures generated by hydraulic fracturing with fracturing fluid, therefore effective conductivity and high output of oil or gas can be obtained [2]. Among the materials used in the hydraulic fracturing, proppant was considered to be the key materials to enhance the exploitation efficiency of oil and gas, especially for the old wells and low permeability reservoir [3].

Proppant is a solid particle with certain sphericity and roundness, strength, corrosion resistance and stability, which can prevent the reclosing of fractures generated by hydraulic fracturing, maintain the smooth flow of oil and gas, and improve the productivity [4,5]. In the hydraulic fracturing process, the shape, strength, acid solubility and turbidity of the proppant can influence the integrity of the newly created fractures, thus determined the efficiency of oil and gas flow out of the well [6]. The sphericity and roundness are very important because it can influence the strength of the proppant. The sphericity is the degree of the proppant close to the sphere, and the roundness refers to the relative sharpness of proppant edges and corners. The angular and pointed proppant tends to break easily, whereas the well-rounded proppant will be mechanically stable. The proppant with high sphericity and roundness can provide high conductivity, because larger passages can be formed for this kind of proppant. The ability to withstand compressive loading is also necessary for proppant. The proppant mustn’t break into fines and block the fractures, which will down the production rate, thus the proppant should with low crushing rate. In addition, the proppant must endure harsh environment such as acid mixture pumped into wells to generated crack, thus the low acid solubility and low turbidity is also very important for the proppant.

Since the natural sand, glass bead and nutshell has been used as proppants in the hydraulic fracturing, new and efficient proppants have been developed rapidly [7]. After 1970s, due to the excellent performance in oil exploitation, synthetic ceramics proppants sintered with bauxite as the main materials was rapid developed and promoted. In order to further improve the comprehensive properties, polymer coated sand and coated ceramic proppant were developed after 1980s [8]. However, traditional natural sand and synthetic ceramics is not suitable for the eco-friendly hydraulic fracturing, because they are settled quickly in low-viscosity fracturing liquid (Figure 2) [9]. In addition, the inertia of these high density proppants makes them hard to turn from the wellbore to perforations [10].

The oil and gas production is highly correlated with the propped length and area, and the fracture conductivity of hydraulic fracturing. As reported in the literature [11], 3D fracturing model shows that ULDP cover higher and longer fracture areas with a smaller propped width. The ULDP can improve the propped length and area in low permeability shale reservoirs, while mainly improve the propped area in high permeability shale reservoirs. On the other hand, the fracture conductivity was highly dependent on the proppant size, flow back, and fines generation [6]. Generally, large proppant size, low flow back and low fines generation can provide higher fracture conductivity.

As the traditional hydraulic fracturing fluid possesses a low viscosity, which has a poor carrying ability for high density proppant, such as natural sand, ceramic and glass bead, which has high density and high fines generation [12]. In order to improve the carrying capacity of traditional proppant, it is necessary to increase the flow rate or viscosity of fracturing fluid. Though these methods can solve the problem of carrying ability to some extent, yet they will cause new problems such as difficult treatment of fluid flow back and great damage to reservoirs [13]. Therefore, in order to thoroughly solve the problem of proppant transport, it is of great significance to develop novel proppant with low density, or new fracturing technology such as liquid self-propping [14]. According to the difference of density, proppants can be divided into following categories (Table 1) [15].

Till now, there are three main strategies to obtain low density proppant: (1) make the traditional HDP into hollow or porous structure, so as to reduce the apparent density of the proppant [5,15]. (2) organic polymer coating for traditional HDP, which can not only reduce the density to some extent, but also can improve the mechanical strength, sphericity, hydrophobicity and corrosion resistance [6,7,8]. (3) develop ULDP with organic polymer, modification with inorganic filler is always needed to improve the mechanical and thermal properties [15]. Though the porosity of traditional HDP can significantly reduce the density, yet the mechanical strength and corrosion resistance will also be affected. Therefore, increasing the content of low density organic polymer in the proppant, including coated proppant and polymer composite proppant, became the most important development direction of high performance LDP. As mentioned above, this review firstly introduced the HDP based on natural sand and synthetic ceramics, then systematically summarized the application of polymer in coated sand, coated ceramic, coated nutshell, especially for the polymer composites based proppant.

## 2. Traditional High Density Proppant

At present, the most used proppant in hydraulic fracturing is high-density natural sand and synthetic ceramics, the typical properties of these proppants are listed in Table 2 [6]. Because of its low cost and relative low density, the natural sand was also widely used in the hydraulic fracturing [7]. However, the natural sand possesses high crushing rate, high turbidity and low sphericity, which is only suitable for the exploitation of shallow oil and gas wells with low closing pressure. With the depth and pressure increase of the oil well, the natural sand can not meet the requirements, high performance synthetic ceramic proppant came into being. Though high-strength ceramic proppant can be made from aluminous minerals and other industrial wastes, yet the fabrication of ceramic proppant needs to be sintered under high temperature, leading to the increase of cost [16,17]. Most of the ceramic proppant was made with high density bauxite as raw materials, leading to the high density of traditional ceramic proppant. To enhance the carrying ability for high density ceramic proppant, the increase of displacement or viscosity for the fracturing fluid is always needed, while this will bring about the problems of difficult treatment of fluid flow back and great damage to oil and gas reservoirs [18].

In order to reduce the density of synthetic ceramic proppants, novel ceramic proppant with hollow or porous structure were developed (Figure 3). Template based method is the most commonly approach to prepare hollow ceramic proppant, and volatile materials are always used as the templates, which can be volatilized through heating in the process of sintering [19]. With the addition of porogen in the fabrication, porous ceramic proppant can be made with the method similar to hollow ceramic proppant [20]. Though the porosity of high density ceramic proppant can significantly reduce the density, the mechanical strength and corrosion resistance will also be affected [15].

## 3. Polymers Used in the Proppant Coating

The coating of traditional proppant with organic polymer can not only decrease the density of the proppants, but also can improve the sphericity and prevent the flow back of proppant (Figure 4) [7]. In addition, the polymer coating can serve as a protective layer to improve the chemical resistance and hydrocarbon affinity, providing interaction between proppant to prevent leakage, as well as wrapping the breaks proppant after the proppant broken [8].

Polymers are obtained through the polycondensation or addition polymerization of small monomers. According to the chemical composition, polymers can be divided into organic polymers and inorganic polymers. According to the thermal behavior, the organic polymers can be divided into thermosetting polymers and thermoplastic polymers. As the thermosetting polymers have many advantages such as easy processing, high strength and low density, they have been widely used in the proppants coating [8]. On the other hand, the thermosetting polymers also possess some disadvantages, including low resistance to oxygen, ozone and radiation, as well as the low thermal stability. The properties of the polymers mainly used in the proppant coating are listed in Table 3 [12], including epoxy resin, phenolic resin, urea-aldehyde, and so on. Among this polymers, epoxy and phenolic resin was the most promising candidate, due to its high performance and low cost.

According to the difference of curing style, the coated proppant can be divided into precuring proppant and curable proppant [7]. The precuring proppant was made by traditional heat curing, individual proppant particle with smoother surface can be obtained. In addition, the compressive strength, acid solubility, sphericity and roundness of the proppant can be improved. For the curable proppant, the polymer was coated on the substrate firstly, then injected into the formed fracture, finally cured and formed network structure under high temperature and pressure in the well. The formed proppant network can not only prevent flow back, but also can decrease the proppant embedding into the clay or shale [8].

## 4. Application of Polymer in Coated Proppant

According to the difference of substrates, the coated proppants can be divided into coated sand, coated ceramic and coated nutshell. Compared with the original substrates, the comprehensive performance of the coated proppants was improved, but the cost will also increase.

### 4.1. Polymer Coated Natural Sand Proppant

Polymer coated sand proppant has been widely investigated and applied (Figure 5), representative research work are shown in Table 4. The effect of polymer type on the comprehensive properties of coated sand was investigated by Zhang and co-workers [21], including furan resin, resole resin, novolak resin and epoxy resin. Compared with natural sand, the properties of coated sand are significantly improved. The crushing rate decreased from 36% to less than 4%, making the coated sand proppant can be used in deep oil wells. The acid solubility decreased to 0.5%, which can greatly improve the resistance to strong acid originated from fracture cleaning fluid. The turbidity decreased from 95 to less than 30, which will reduce the pollution of proppant to oil and soil. The sphericity and roundness increase from 0.6 to 0.8, which will enhance the conductivity. Carbon nanotube (CNT) toughened epoxy coated sand was reported by Xia [22], high performance coated sand proppant was obtained with obvious improved conductivity. Epoxy/phenolic binary mixture coated sand was reported in our previous work [23], the sphericity and roundness of the obtained proppant was further enhanced to 0.85. Multicomponent polymer (polystyrene, polymethyl methacrylate, epoxy) coated sand was investigated by Krishnan and co-workers [24,25], ultra-low density proppant with bulk density less than 1.5 g·cm^−3^ was obtained, but the improvement of crushing rate and sphericity is not obvious.

The coating of sand can not only improve the intrinsic characteristics, but also can address the application problems in oil exploitation. Poly(2-fluorine-4-vinylpyridine) (PFVP) coated sand was developed for the hydraulic fracturing, the crack conductivity and flow back control ability of the obtained proppant was significantly improved [26]. To give a deep insight into the chemical stability of the coated sand, novolac resin coated sand was fabricated and tested at different conditions [27], results revealed that the leaching behavior under high temperature (400 °C) and strong acid (pH = 3) was improved. In addition, the concentration of phenolics in the aqueous leachates are lower than 1 ppm. Recently, graphene oxide (GO)/silane coupling agent (KHS) coated sand was also reported [28], though the comprehensive performance was not obviously improved, yet the fabrication method is relative simple.

### 4.2. Polymer Coated Synthetic Ceramic Proppant

Polymer coated ceramic proppant has also been widely investigated and applied (Figure 6), representative research work are shown in Table 5. Deng and co-workers reported the fabrication of ceramic proppant, which sintered with industrial waste fly ash, maganese oxide and potassium feldspar [29]. After coated with phenolic/epoxy binary mixture, the apparent density decreased from 3.11 to 2.64 g·cm^−3^, the crushing rate has also reduced from 20.3 to 3.76%. Low-cost ceramic proppant was synthesized with kaolin and ZnO [30], the apparent density further decreased to 2.27 g·cm^−3^ with epoxy coating.

To further reduce the apparent density, coated porous and hollow ceramic proppant was developed. Phenolic-epoxy coated porous mullite based ceramic proppant was reported by Guo and co-workers [31], the apparent density further decreased to 1.90 g·cm^−3^, the compressive strength, sphericity and roundness of the obtained proppant was also improved. Hollow ceramic proppant was fabricated with silica fume as raw material and urea as porogen [32], epoxy coated ULDP with apparent density of 1.03 g·cm^−3^ was obtained for this proppant. Due to its hollow structure, the crushing rate increased to 17.01% at 55 MPa.

The influence of curing conditions on the performance of phenolic resin coated ceramic proppant was deeply investigated by Zhang and co-workers [33]. The obtained results revealed that high temperature is conducive to forming a coating with excellent barrier properties, and low temperature is benefit for improving the sphericity and roundness. As the hydrophobicity of the proppant has an important influence on the recovery efficiency of hydraulic fracturing, which can reduce the adsorption of water and increase the conductivity of oil and gas, therefore it is necessary to enhance the hydrophobicity of the proppant [6,8,12]. The practical application of epoxy coated ceramic proppant was investigated by Fan and co-workers [34], the self-suspension ability of the coated proppant increased by 16 times, the hydrophobicity and conductivity were increased by 83.8% and 16.71%, respectively. ULW-1.75 is the most popular commercial coated ceramic proppant [35,36], which has an average porosity of 50% and can form a ULDP with apparent density of 1.75 g·cm^−3^. In addition, the closing pressure of 56 MPa can be tolerated at 121 °C.

### 4.3. Polymer Coated Bio-Based Nutshell Proppant

As the nutshell possesses ultra-low density, relative high strength and low cost, leading to the widely application since 1960s [37]. In order to give a deep insight into the performance of raw nutshell used directly in hydraulic fracturing, the comprehensive properties of coconut shell, palm shell and walnut shell was systematically investigated by Zoveidavianpoor and co-workers [38] (Table 6). The apparent density of these shells in the range of 1.14 to 1.33 g·cm^−3^, but the crushing rate, acid solubility and turbidity is relative poor. To address the problem of poor corrosion and heat resistance, especially the low sphericity and roundness for raw nutshell, coated nutshell proppant came into being (Figure 7). On the basis of their previous work, Zoveidavianpoor investigated the properties of epoxy coated coconut shell proppant [39]. The crushing rate of the obtained coated proppant decreased from 2.12 to 0.16%, the acid solubility and turbidity reduced to 1.8 and 38 respectively, the sphericity and roundness increased from 0.7 to 0.8.

In the investigation of coated nutshell, nutshell particles with 20~40 mesh was always used. Phenolic resin coated nutshell particles was reported by Li and co-workers [40], the obtained proppant possesses an apparent density of 1.23 g·cm^−3^, the water adsorption decreased from 30 to 17%, and the conductivity increased from 41.8 to 113.4 μm^2^·cm. Li and co-workers [41] has also further investigated the coating times on the properties of coated nutshell, the properties of water adsorption and compression deformation were improved, the sphericity and roundness increased to 0.86. Coated nutshell with phenolic resin impregnation and epoxy resin coating was reported by Huang [42,43], no obvious crack was observed under 60 MPa, and the water adsorption decreased from 30.45 to 6.58%.

ULW-1.25 is the most popular commercial coated nutshell proppant [35,36], after polymer coating the apparent density increased to 1.25 g·cm^−3^, the sphericity and roundness increased from 0.5 to 0.62, and the closing pressure of 42 MPa can be tolerated at 79 °C [11,44]. The simulation study shows that at the same injection concentration in slickwater, the ULW-1.25 creates more propped area and longer propped distance than natural sand, but the fracture conductivity is lower.

## 5. Polymer Composites Based Ultra-Low Density Proppant

Since the organic polymer has a density close or even lower than that of water, polymer composites became the most promising ULDP for clean hydraulic fracturing. About two decades ago, polymer composite based ULW-1.05 proppant was applied in the hydraulic fracturing [11,44], which is a heat-treated nanopolymer microsphere with an apparent density of 1.05 g·cm^−3^. The sphericity and roundness of ULW-1.05 is greater than 0.9, the acid solubility is less than 2%, the glass transition temperature of approximately 145 °C, the closing pressure of 55 MPa can be tolerated at 130 °C [13]. The disadvantage of ULW-1.05 is that it is prone to deformation compared to traditional fracturing proppants, this can be ascribed to the intrinsic elasticity of polymer.

As we all know, ultra-low density cross-linked polymer beads with very high sphericity can be obtained with suspension polymerization, and this technology has been widely used in the industry, such as the scale production of ion exchange resin [45] and chromatographic column packing [46]. In addition, the size of the proppant prepared with suspension polymerization can be facile controlled by the stirring speed and dispersant concentration [45,46]. This will benefit for the enhancing of conductivity in deep reservoirs, where small-size proppants are preferred since they can flow into narrower fractures (Figure 8) [9].

What’s more, the comprehensive properties of cross-linked polymer composite beads obtained from suspension polymerization can be improved by further chemical cross-linking and incorporation with inorganic fillers. Therefore, proppant based on polystyrene microspheres composite was reported by our group for the first time [47,48]. In the study, graphite/polystyrene composite microspheres were prepared through suspension polymerization, and the properties for proppant application were investigated, including density, crushing rate, sphericity and heat resistance. Now, the ULDP originated from polymer composite based suspension polymerization has been widely studied and applied (Figure 9).

### 5.1. Ultra-Low Density Proppant Based on Polystyrene

Since polystyrene (PS) has excellent compressive strength and wear resistance, cross-linked PS microspheres have been widely studied and applied in ULDP (Table 7). In order to reveal the performance of pure PS microspheres directly used as ULDP, pure PS microspheres was synthesized through suspension polymerization using styrene as monomer and divinylbenzene as cross-linking [49]. The obtained pure PS microspheres has an apparent density as low as 1.03 g·cm^−3^, the crushing rate is 3.01% under 69 MPa, and the sphericity is greater than 0.9. In order to further improve the comprehensive performance of pure PS proppant, inorganic filler filled composites microspheres proppant has been developed rapidly. In our previous work, graphite/PS composite microspheres was prepared through suspension polymerization [47]. Novel ULPD was obtained with a low density of 1.05 g·cm^−3^, a crushing rate significantly reduced to 1.3%, a decomposition temperature of 5% reached to 325 °C. Simultaneously, SiO_2_/PS composite microspheres was prepared with increased apparent density of 1.07 g·cm^−3^, the decomposition temperature of 5% increased to 372 °C [48]. In addition, PS composite with incorporation of low cost carbon black (CB) was reported [50], the obtained proppant possesses an apparent density of 1.08 g·cm^−3^. Due to its excellent properties [51,52], graphene incorporated PS composite microspheres was also investigated in our previous work [53], the performance of the obtained proppant is similar to ULW-1.05. Carbon nanotube incorporated PS composite microspheres was reported by Tasque and co-workers [54], the crushing rate as low as 1.7% under 138 MPa, and the acid solubility is also very low.

Silica fume (SF)/PS composite microspheres was also investigated in our previous work [55], the obtained proppant has low acid solubility of 1.75%, and high contact angle of 124.5°. In addition, commercial strong acid PS cation exchange resin has also been used for the preparation of ULDP [56], which has been cross-linked with heavy metal cations such as iron ion and barium ion, the apparent density of the obtained proppant increased to 1.56 g·cm^−3^, and the crushing rate decreased to 0.6% at 52 MPa. Based on the graphite/PS proppant [48], coated ULDP proppant was prepared with coating of epoxy/phenolic resin [57,58]. The density of the obtained coated proppant increased from 1.05 to 1.08 g·cm^−3^, and the acid solubility decreased to 0.11%.

### 5.2. Ultra-Low Density Proppant Based on Polymethyl Methacrylate

Since polymethyl methacrylate (PMMA) has high modulus and excellent chemical resistance, cross-linked PMMA microspheres have also been widely studied and applied in ULDP (Table 8). In the investigation of Huang and co-workers [59], silane coupling agent A151 (vinyl triethoxysilane) was added in the preparation of cross-linked PMMA microspheres with suspension polymerization. The A151 can not only covalent linked to the cross-linked PMMA through the addition polymerization of double bond, but also can further cross-linked with the hydrosis and polycondensation of ethoxy, which can form a strong inorganic coating on the surface. In another report, pine bark (PB) was used for the preparation of ULDP with low cost [60], the obtained proppant has an apparent density as low as 1.02 g·cm^−3^, and only slight deformation was observed within 5 min under 69 MPa.

Recently, the influence of inorganic fillers such as graphite (G) [61], carbon black (CB) [62], silica fume (SF) [63] and fly ash (FA) [64] on the performance of PMMA based ULDP has been deeply investigated by our group. All of the obtained ULDP has an apparent density less than 1.25 g·cm^−3^, the crushing rate is no more than 3% and the sphericity is greater than 0.9, as well as excellent corrosion resistance and low turbidity.

As shown in Figure 10, the PMMA composited of polar aliphatic chains, which possesses good flexibility and strong intermolecular interaction, leading to the high modulus and low abrasion resistance of PMMA [65,66]. On the other hand, the PS composited of nonpolar aromatic chains, which possesses rigid structure and weak intermolecular interaction, making the PS has a low modulus and high abrasion resistance [67,68]. In order to make up for the shortcomings of homopolymer [69,70,71], ULDP based on FA incorporated poly(St-*co*-MMA) composite microsphere was synthesized and characterized (Figure 11) [72]. In addition, ULDP based on acrylonitrile (AN) and vinyl acetate (VA) copolymer microsphere has also been reported, the obtained proppant has a good self-suspension ability [9].

**Table 8 polymers-14-05534-t008:** Properties of PMMA based ULDP.

Samples	Apparent Density (g·cm^−3^)	Crushing Rate (%)	Acid Solubility (%)	Turbidity (FTU)	Sphericity and Roundness	Ref.
A151/PMMA	1.08	6.3/69 MPa	/	/	>0.9	[59]
PB/PMMA	1.02	5 min/69 MPa	/	/	>0.9	[60]
G/PMMA	1.06	3.0/69 MPa	0.08	35	>0.9	[61]
CB/PMMA	1.23	2.44/69 MPa	0.08	33	>0.9	[62]
SF/PMMA	1.16	2.2/69 MPa	2	/	>0.9	[63]
FA/PMMA	1.05	3.0/69 MPa	0.02	54	>0.9	[64]
FA/P(St-*co*-MMA)	1.26	0.95/52 MPa	/	/	>0.9	[72]
P(AN-*co*-VA)	1.04	0.2/52 MPa	<1	/	>0.9	[9]

## 6. Conclusions and Future Prospects

Proppant is the key materials for hydraulic fracturing, with the increase of low permeability reservoir and the high requirements of environment protection, the demand for LDP is growing rapidly, especially for the ULDP. However, the density of traditional natural sand and synthetic ceramic proppant is too high, and the properties of sphericity, crushing rate, acid solubility and turbidity is not ideal. Though the coating of high density proppant can reduce the density to some extent, yet the obtained coated sand and coated ceramic only can be regarded as LDP.

In order to obtain ULDP, organic matrix based materials was rapidly developed for its construction, including nutshell and polymer composites. Though the nutshell based proppant is renewable and cheap, yet the sphericity and compressive strength is very low, the properties of acid solubility, turbidity and heat resistance is also not ideal. In addition, the coating of nutshell also has limited effect on improving the performance. Based on suspension polymerization, ultra-low density (less than 1.25 g·cm^−3^) polymer composite microsphere with very high sphericity (more than 0.9) can be obtained. In addition, the compressive strength and heat resistance of polymer composites based proppant can be improved by chemical cross-linking and inorganic filler incorporation. Therefore, polymer composite based ULDP will be the most promising candidate.

Now, PS and PMMA composite microsphere based ULDP has been widely investigated. With the development of polymer composites, besides the low density and high sphericity, polymer composite microsphere with improved crushing rate, acid solubility, turbidity and heat resistance has also been reported, which has good application prospect in the field of hydraulic fracturing. However, there are also some problems need to be addressed in the investigation and application. Firstly, there are few researches about the practical application performance of this kind of ULDP, especially for the conductivity. Secondly, the cost of polymer materials is higher than that of sand, ceramic and nutshell. It is hoped that these problems can be solved in the future research, so the industrial application of polymer composites base ULDP can be realized.

## Figures and Tables

**Figure 1 polymers-14-05534-f001:**
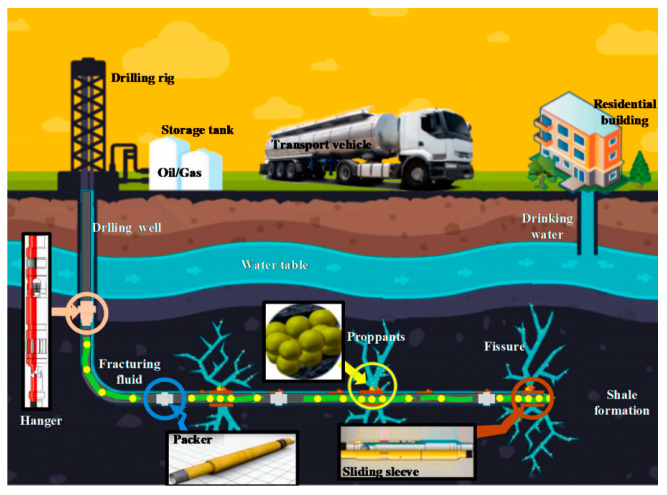
Schematic of the hydraulic fracturing process [1].

**Figure 2 polymers-14-05534-f002:**
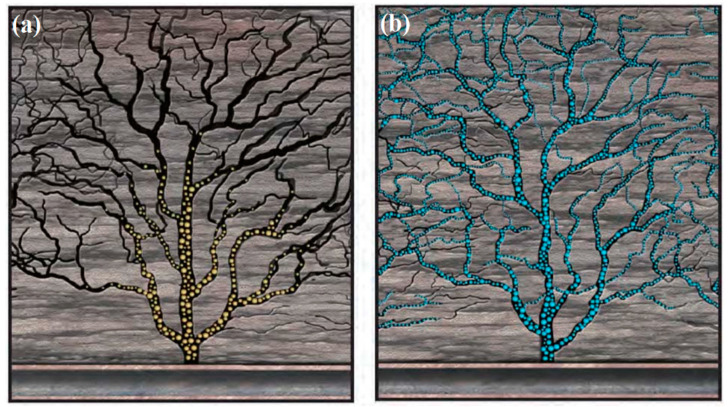
Prop effect of (**a**) traditional proppant and (**b**) low-density proppant [9].

**Figure 3 polymers-14-05534-f003:**
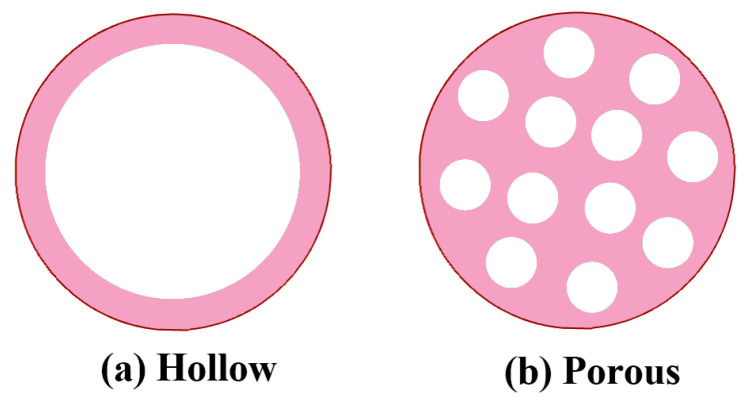
Schemes of the hollow (**a**) or porous (**b**) ceramic proppant.

**Figure 4 polymers-14-05534-f004:**
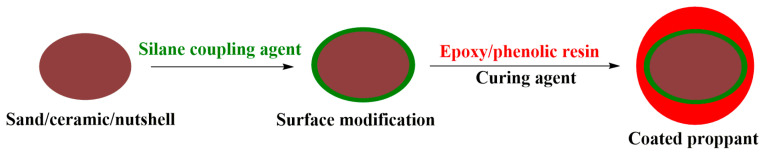
Schemes of the coating process of traditional proppant.

**Figure 5 polymers-14-05534-f005:**
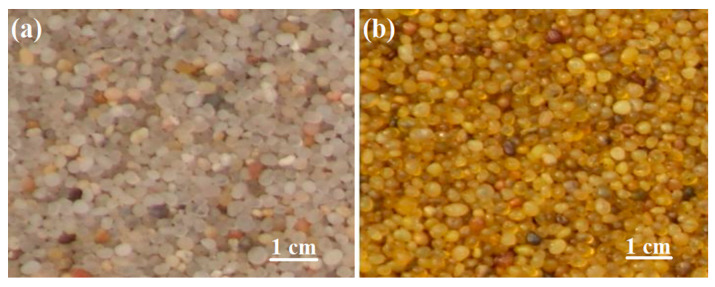
Sand (**a**) and coated sand (**b**) proppant.

**Figure 6 polymers-14-05534-f006:**
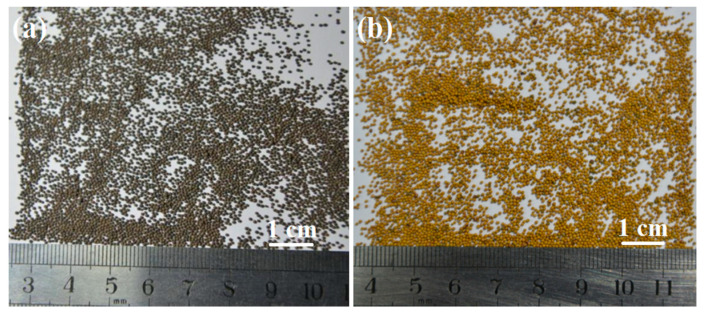
Ceramic (**a**) and coated ceramic (**b**) proppant.

**Figure 7 polymers-14-05534-f007:**
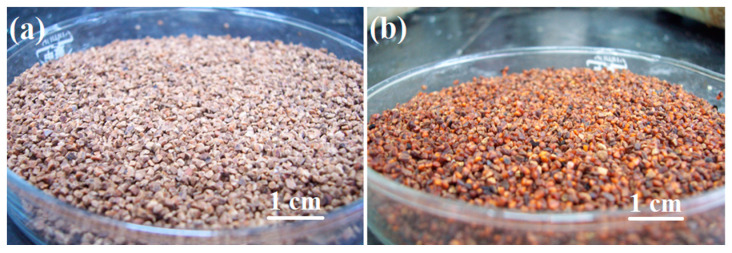
Nutshell (**a**) and coated nutshell (**b**) proppant.

**Figure 8 polymers-14-05534-f008:**
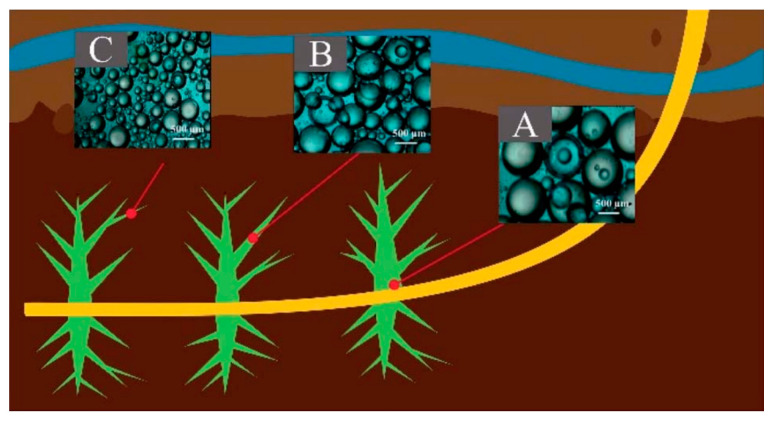
Hydraulic fracture propped with proppant of (**A**) large, (**B**) medium, (**C**) small size [9].

**Figure 9 polymers-14-05534-f009:**
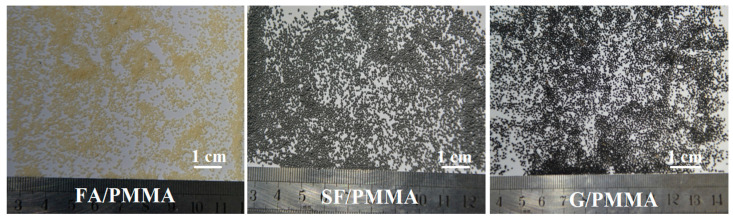
ULDP based on polymer composite beads.

**Figure 10 polymers-14-05534-f010:**
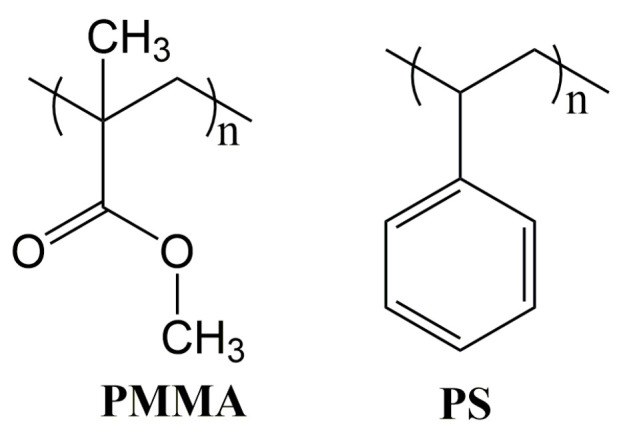
Chemical structure of PMMA and PS.

**Figure 11 polymers-14-05534-f011:**
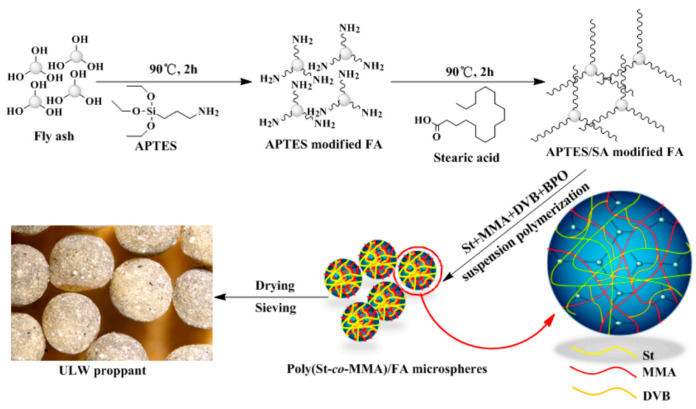
Schemes of the synthetic process of poly(St-*co*-MMA)/FA composite proppant [72].

**Table 1 polymers-14-05534-t001:** Proppant classification according to the density.

Proppant	Bulk Density (g·cm^−3^)	Apparent Density (g·cm^−3^)
High density (HDP)	>1.8	>3.35
Intermediate density (IDP)	1.65~1.80	3.00~3.35
Low density (LDP)	1.50~16.5	2.60~3.00
Ultra-low density (ULDP)	<1.5	<2.6

**Table 2 polymers-14-05534-t002:** Typical properties of natural sand and synthetic ceramic proppant.

Proppant	Natural Sand	Synthetic Ceramics
Apparent density (g·cm^−3^)	2.5~2.7	3.3~3.6
Crushing rate (%)	36	13.5
Acid solubility (%)	5.5	6.9
Turbidity (FTU)	95	60
Sphericity	0.6	0.8
Roundness	0.6	0.8
Cost	low	high

**Table 3 polymers-14-05534-t003:** Properties for the polymers used in the proppant coating.

Polymer	Curing Temperature °C	Strength	Acid Resistance	Heat Resistance	Hydrophobic	Chemical Resistance
Epoxy resin	66~204	good	good	very good	good	good
Phenolic-aldehyde	121~204	good	good	very good	good	good
Urea-aldehyde	121~204	good	good	very good	good	good
Polyurethane	99~121	good	moderate	good	good	moderate
Vinyl resin	100~149	moderate	good	moderate	moderate	moderate
Polyester	100~149	moderate	moderate	moderate	moderate	moderate
Furan resin	191	bad	good	moderate	moderate	good

**Table 4 polymers-14-05534-t004:** Properties of polymer coated natural sand proppant.

Polymer	Bulk Density (g·cm^−3^)	Crushing Rate (%)	Acid Solubility (%)	Turbidity (FTU)	Sphericity and Roundness	Ref.
Furan resin	1.58	13/69 MPa	0.49	30	0.8	[21]
Resole resin	1.57	3.9/69 MPa	0.51	26	0.8	
Novolak resin	1.56	3.8/69 MPa	0.50	28	0.8	
Epoxy resin	1.58	1.9/69 MPa	0.18	24	0.8	
CNT-Epoxy	/	5.21/69 MPa	2.83	24	0.8	[22]
Epoxy/Phenolic	/	1.6/52 MPa	/	/	0.85	[23]
(PS-PMMA-CG)- (epoxy-CG)	1.35	9/69 MPa	/	/	0.7	[24]
(PS-PMMA/ DVB)-(Epoxy-CG)	1.32	5/82 MPa	/	/	0.7	[25]
PFVP	The conductivity and flow back control ability was significantly improved					[26]
Novolac resin	Leaching behavior under high temperature and strong acid was improved					[27]
GO-KHS	1.58	20/10 MPa	0.7	55	0.8	[28]

**Table 5 polymers-14-05534-t005:** Properties of polymer coated synthetic ceramic proppant.

Polymer	Apparent Density (g·cm^−3^)	Crushing Rate (%)	Acid Solubility (%)	Turbidity (FTU)	Sphericity and Roundness	Ref.
Phenolic/Epoxy	2.64	3.76/52 MPa	/	/	/	[29]
Epoxy	2.27	1.16/69 MPa	/	/	/	[30]
Phenolic-Epoxy	1.90	2.81/69 MPa	/	/	>0.9	[31]
Epoxy	1.03	17.01/55 MPa	/	1.59	>0.9	[32]
Phenolic	/	18/82 MPa	4	/	0.9	[33]
Epoxy	/	Hydrophobicity, suspension ability and conductivity improved				[34]
ULW-1.75	1.75	Average porosity 50%, 56 MPa can be tolerated at 121 °C				[35,36]

**Table 6 polymers-14-05534-t006:** Properties of polymer coated bio-based nutshell proppant.

Polymer	Apparent Density (g·cm^−3^)	Crushing Rate (%)	Acid Solubility (%)	Turbidity (FTU)	Sphericity and Roundness	Ref.
Coconut shell *	1.22–1.33	2.12/55 MPa	44	184	0.7	[38]
Palm shell *	1.16–1.19	4.22/55 MPa	52	190	0.7	
Walnut shell *	1.14–1.25	7.0/55 MPa	47	188	0.7	
Epoxy	/	0.16/55 MPa	1.8	38	0.8	[39]
Phenolic	1.23	15/60 MPa	Conductivity increased from 41.8 to 113.4 μm^2^·cm			[40]
Phenolic	1.24	14.8/60 MPa	/	/	0.86	[41]
Phenolic/Epoxy	/	No crack under 60 MPa, water adsorption significantly reduced				[42,43]
ULW-1.25	1.25	42 MPa can be tolerated at 79 °C				[35,36]
ULW-1.25	1.25	Conductivity ability was obviously improved			0.62	[11,44]

* Raw materials of nutshell.

**Table 7 polymers-14-05534-t007:** Properties of PS based ULDP.

Samples	Apparent Density (g·cm^−3^)	Crushing Rate (%)	Acid Solubility (%)	Sphericity and Roundness	Ref.
PS	1.03	3.01/69 MPa	/	>0.9	[49]
G/PS	1.05	1.3/69 MPa	/	0.94	[47]
SiO_2_/PS	1.07	3.0/69 MPa	/	0.95	[48]
CB/PS	1.08	2.0/69 MPa	/	>0.9	[50]
Graphene/PS	1.05	0.45/52 MPa	/	0.95	[53]
CNT/PS	1.05	1.7/138 MPa	0.9	0.9	[54]
SF/PS	1.05	1.0/52 MPa	1.75	>0.9	[55]
Metal/PS	1.56	0.6/52 MPa	/	>0.9	[56]
Coated-G/PS	1.08	2.22/69 MPa	0.11	>0.9	[48,57,58]

## Data Availability

Not applicable.

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
