# Peer review of "Progress of Polymer Application in Coated Proppant and Ultra-Low Density Proppant"

_polymers, 2022, doi:10.3390/polym14245534_

Round 1

Reviewer 1 Report

This paper reviewed several different types of proppants based on their density and material. The authors discussed the advantages and disadvantages of different proppants. This paper focused on reviewing how the polymer can be used to fabricate low density and ultra-low density proppants. Some general benefits of using polymer include reducing proppant density, enhancing mechanical strength, and increasing hydrophobicity of proppant surface.  

My comments and suggestions are summarized below.

  1. The paper title is not very accurate. The authors should consider adding Ultra-low density proppant into the title since this paper has a large portion on ULDP.
  2. The authors should add some visual aids such as illustration figures and charts to help readers understand this paper. Some improvements include but are not limited to:
    1. Illustration figures of hydraulic fracturing and propped fractures in subsurface reservoirs.
    2. Illustration figures of different proppants: sand, ceramic, nutshell, polymer coated proppant, and polymer composite proppant.
    3. Illustration figures of hollow or porous proppants.
    4. Flowcharts or illustrations of some proppant manufacturing processes for hollow proppants, coated proppants, and polymer composite proppants (suspension polymerization).
  3. In the Introduction section, the authors should elaborate more on the connections between proppant properties and oil production. It will help readers understand how different proppant properties affect oil production, and what kind of proppant properties are preferred. Some engineering implications include but are not limited to:
    1. The oil production is highly correlated with the overall propped area and length of hydraulic fractures. The authors should consider discussing how the proppant density affects the propped area and length, and why low density or ultra-low density is desired from the oil production point of view.
    2. Oil production also depends on fracture conductivity under the reservoir closure stress. The authors should consider discussing what proppant properties help increase and maintain fracture conductivity. This will help explain why introducing polymer to proppant manufacturing.
    3. The authors mentioned hydrophobicity and contact angle data for some types of proppants. But the authors did not discuss the effect of proppant surface wettability on the oil flow and production. The authors should consider discussing what kind of proppant wettability is preferred and adding related references.
  4. The authors briefly compared PS with PMMA in lines 275-278. The authors should consider showing the molecular structures of these two different polymers and explaining why they have different properties in a more mechanistic way.
  5. The proppant size is also a very important property. Especially in deep reservoirs, the width of hydraulic fracture tends to be smaller than that in shallow reservoirs due to higher in-situ stress. Sometimes the small-size proppants are preferred since they can flow into narrower hydraulic fractures. It will be a good addition if the authors can discuss the implications and challenges of introducing polymer for controlling proppant size, especially the small size during the fabrication process.
  6. What is the difference between the sphericity and roundness of proppants? If these two concepts are identical, the authors should consider using only one word to avoid further confusion. If they are different, the authors should consider explaining the differences.
  7. There are several typos and grammar errors in this paper. The authors should improve their English writing. The identified typos and errors include but are not limited to:
    1. Line 28: prappant -> proppant
    2. Line 44: carring -> transport
    3. Line 53: need -> needed
    4. Line 55, 84: strengthen -> strength
    5. Line 64: preperties -> properties
    6. Line 74: "This cause the displacement or viscosity of fracturing fluid needs to be increased" should be rephrased to be grammatically correct.
    7. Line 91: broken -> breaks
    8. Line 111: rejected -> injected
    9. Line 114: soil -> clay or shale
    10. Line 128: decrease -> decreased
    11. Line 129: reduced -> reduce
    12. Line 144: test -> tested
    13. Line 148: obvious improve -> obviously improved
    14. Line 250: 1.75 -> 1.75%
    15. Line 291: obtained -> obtain
    16. Line 293: spherricity -> sphericity

Author Response

Response to the Reviewer 1 

This paper reviewed several different types of proppants based on their density and material. The authors discussed the advantages and disadvantages of different proppants. This paper focused on reviewing how the polymer can be used to fabricate low density and ultra-low density proppants. Some general benefits of using polymer include reducing proppant density, enhancing mechanical strength, and increasing hydrophobicity of proppant surface.  

My comments and suggestions are summarized below.

  • . The paper title is not very accurate. The authors should consider adding Ultra-low density proppant into the title since this paper has a large portion on ULDP.

Response: Thanks for the reviewer’s valuable suggestions. According to the reviewer’s suggestion, the title has been revised to “Progress of Polymer Application in Coated Proppant and Ultra-Low Density Proppant”.

  • . The authors should add some visual aids such as illustration figures and charts to help readers understand this paper. Some improvements include but are not limited to:

(a)Illustration figures of hydraulic fracturing and propped fractures in subsurface reservoirs.

(b)Illustration figures of different proppants: sand, ceramic, nutshell, polymer coated proppant, and polymer composite proppant.

(c)Illustration figures of hollow or porous proppants.

(d)Flowcharts or illustrations of some proppant manufacturing processes for hollow proppants, coated proppants, and polymer composite proppants (suspension polymerization).

Response: Thanks for the reviewer’s valuable suggestion. Eight visual tables were present in the original manuscript, and eleven figures about the hydraulic fracturing and propped fractures, different proppants and the manufacturing processes was added to help readers understand this paper, and the whole manuscript was revised according to the added Figure. Limit to the available resource and the permission of copyright, the illustrations of manufacturing processes for hollow and porous proppants can’t provided.

In the revised manuscript:

Figure 1. Schematic of the hydraulic fracturing process [1].

Figure 2. Prop effect of (a) traditional proppant and (b) low-density proppant [9].

Figure 3. Schemes of the hollow or porous ceramic proppant.

Figure 4. Schemes of the coating process of traditional proppant.

Figure 5. Sand (a) and coated sand (b) proppant. Adapted from Gao et al. Unpulishied.

Figure 6. Ceramic (a) and coated ceramic (b) proppant. Adapted from Gao et al. Unpulishied.

Figure 7. Nutshell (a) and coated nutshell (b) proppant. Adapted from Gao et al. Unpulishied.

Figure 8. Hydraulic fracture propped with proppant of different size [9].

Figure 9. ULDP based on polymer composite beads. Adapted from Gao et al. Unpulishied.

Figure 10. Chemical structure of PMMA and PS.

Figure 11. Schemes of the synthetic process of poly(St-co-MMA)/FA composite proppant [72].

  • . In the Introduction section, the authors should elaborate more on the connections between proppant properties and oil production. It will help readers understand how different proppant properties affect oil production, and what kind of proppant properties are preferred. Some engineering implications include but are not limited to:

Response: Thanks for the reviewer’s valuable suggestion, the connections between proppant properties and oil production was discussed in the revised manuscript.

In the revised manuscript:

Proppant is a solid particle with certain sphericity and roundness, strength, corrosion and stability, which can prevent the reclosing of fractures generated by hydraulic fracturing, maintain the smooth flow of oil and gas, and improve the productivity [4,5]. In the hydraulic fracturing process, the shape, strength, acid solubility and turbidity can influence the integrity of the newly created fractures, thus determined the efficiency of oil and gas flow out of the well [6]. The sphericity and roundness are very important because it can influence the strength of the proppant. The sphericity is the degree of the proppant close to the sphere, and the roundness refers to the relative sharpness of proppant edges and corners. The angular and pointed proppant tends to break easily, whereas the well-rounded proppant will be mechanically stable. The proppant possesses high sphericity and roundness can provide high conductivity, because larger passages can be formed for this kind of proppant. The ability to withstand compressive loading is also necessary for proppant. The proppant mustn’t break into fines and block the fractures, and slowing down the production rate, thus the proppant should possesses low crushing rate. In addition, the proppant must endure harsh environment such as acid mixture pumped into wells to generated crack, thus the low acid solubility and low turbidity is also very important for the proppant.

(a)The oil production is highly correlated with the overall propped area and length of hydraulic fractures. The authors should consider discussing how the proppant density affects the propped area and length, and why low density or ultra-low density is desired from the oil production point of view.

Response: Thanks for the reviewer’s valuable suggestion, the effect of proppant density on the propped area and length was discussed in the revised manuscript.

In the revised manuscript:

Since the natural sand, glass bead and nutshell has been used as proppants in the hydraulic fracturing, new and efficient proppants have been developed rapidly [7]. After 1970s, due to the excellent performance in oil exploitation, synthetic ceramics proppants sintered with bauxite as the main materials was rapid developed and promoted. In order to further improve the comprehensive properties, polymer coated sand and coated ceramic proppant were developed after 1980s [8]. However, traditional natural sand and synthetic ceramics is not suitable for the eco-friendly hydraulic fracturing, because they are settle quickly in low-viscosity fracturing liquid (Figure 2) [9]. In addition, the inertia of these high density proppants makes them hard to turn from the wellbore to perforations [10].

The oil and gas production is highly correlated with the propped length and area, and the fracture conductivity of hydraulic fracturing. As reported in the literature [11], 3D fracturing model shows that ULDP cover higher and longer fracture areas with a smaller propped width. The ULDP can improve the propped length and area in low permeability shale reservoirs, while mainly improve the propped area in high permeability shale reservoirs. On the other hand, the fracture conductivity was highly dependent on the proppant size, flow back, and fines generation [6]. Generally, large proppant size, low flow back and low fines generation can provide higher fracture conductivity.

As the traditional hydraulic fracturing fluid possesses a low viscosity, which has a poor carrying ability for high density proppant, such as natural sand, ceramic and glass bead, which has high density and high fines generation [12]. In order to improve the carrying capacity of traditional proppant, it is necessary to increase the displacement or viscosity of fracturing fluid. Though these methods can solve the problem of carrying ability to some extent, yet they will cause new problems such as difficult treatment of fluid flow back and great damage to reservoirs [13]. Therefore, in order to thoroughly solve the problem of proppant transport, it is of great significance to develop novel proppant with low density, or new fracturing technology such as liquid self-propping [14]. According to the difference of density, proppants can be divided into following categories (Table 1) [15].

(b)Oil production also depends on fracture conductivity under the reservoir closure stress. The authors should consider discussing what proppant properties help increase and maintain fracture conductivity. This will help explain why introducing polymer to proppant manufacturing.

Response: Thanks for the reviewer’s valuable suggestion, the effect of proppant properties on the fracture conductivity was discussed in the revised manuscript.

In the revised manuscript:

The oil and gas production is highly correlated with the propped length and area, and the fracture conductivity of hydraulic fracturing. As reported in the literature [11], 3D fracturing model shows that ULDP cover higher and longer fracture areas with a smaller propped width. The ULDP can improve the propped length and area in low permeability shale reservoirs, while mainly improve the propped area in high permeability shale reservoirs. On the other hand, the fracture conductivity was highly dependent on the proppant size, flow back, and fines generation [6]. Generally, large proppant size, low flow back and low fines generation can provide higher fracture conductivity.

(c)The authors mentioned hydrophobicity and contact angle data for some types of proppants. But the authors did not discuss the effect of proppant surface wettability on the oil flow and production. The authors should consider discussing what kind of proppant wettability is preferred and adding related references.

Response: Thanks for the reviewer’s valuable suggestion, the effect of proppant surface wettability on the oil flow and production was discussed in the revised manuscript.

In the revised manuscript:

The influence of curing conditions on the performance of phenolic resin coated ceramic proppant was deeply investigated by Zhang and co-workers [33]. The obtained results revealed that high temperature is conducive to forming a coating with excellent barrier properties, and low temperature is benefit for improving the sphericity and roundness. As the hydrophobicity of the proppant has an important influence on the recovery efficiency of hydraulic fracturing, which can reduce the adsorption of water and increase the conductivity of oil and gas, therefore it is necessary to enhance the enhance the hydrophobicity of the proppant [6,8,12]. The practical application of epoxy coated ceramic proppant was investigated by Fan and co-workers [34], the self-suspension ability of the coated proppant increased by 16 times, the hydrophobicity and conductivity were increased by 83.8 % and 16.71%, respectively. ULW-1.75 is the most popular commercial coated ceramic proppant [35,36], which has an average porosity of 50% and can form a ULDP with apparent density of 1.75 g·cm-3. In addition, the closing pressure of 56 MPa can be tolerated at 121 °C.

Ref.:

[6] Michael, F. M.; Krishnan, M. R.; Li, W. G.; Alsharaeh, E. H. A review on polymer-nanofiller composites in developing coated sand proppants for hydraulic fracturing. J. Nat. Gas Sci. Eng. 2020, 83, 103553. DOI: 10.1016/j.jngse.2020.103553.

[8] Pangilinan, K. D.; Leon, A. C.; Advincula, R. C. Polymers for proppants used in hydraulic fracturing. J. Petrol. Sci. Eng. 2016, 145, 154-160. DOI: 10.1016/ j.petrol.2016.03.022.

[12] Zoveidavianpoor, M.; Gharibi, A. Application of polymers for coating of proppant in hydraulic fracturing of subterraneous formations: A comprehensive review. J. Nat. Gas Sci. Eng. 2015, 24, 197-209. DOI: 10.1016/j.jngse.2015.03.024.

[34] Fan, F.; Li, F. X.; Tian, S. C.; Sheng, M.; Khan, W.; Shi, A. P.; Zhou, Y.; Xu, Q. Hydrophobic epoxy resin coated proppants with ultra-high self suspension ability and enhanced liquid conductivity. Petrol. Sci. 2021, 18, 1753-1759. DOI: 10.1016/j.petsci. 2021.09.004.

  • . The authors briefly compared PS with PMMA in lines 275-278. The authors should consider showing the molecular structures of these two different polymers and explaining why they have different properties in a more mechanistic way.

Response: Thanks for the reviewer’s valuable suggestion, the chemical structure of PMMA and PS was provided, and the different properties was explained according to the difference of the structure.

In the revised manuscript:

Figure 10. Chemical structure of PMMA and PS.

As shown in Figure 10, the PMMA composited of polar aliphatic chains, which possesses good flexibility and strong intermolecular interaction, leading to the high modulus and low abrasion resistance of PMMA [65,66]. On the other hand, the PS composited of nonpolar aromatic chains, which possesses rigid structure and weak intermolecular interaction, making the PS has a low modulus and high abrasion resistance [67,68]. In order to make up for the shortcomings of homopolymer [69-71], ULDP based on FA incorporated poly(St-co-MMA) composite microsphere was synthesized and characterized (Figure 11) [72]. In addition, ULDP based on acrylonitrile (AN) and vinyl acetate (VA) copolymer microsphere has also been reported, the obtained proppant has a good self-suspension ability [9].

Ref.:

  • Blond, D.; Barron, V.; Ruether, M.; Ryan, K. P.; Nicolosi, V.; Blau, W. J.; Coleman, J. N. Enhancement of modulus, strength, and toughness in poly(methyl methacrylate)-based composites by the incorporation of poly(methyl methacrylate)- functionalized nanotubes. Adv. Funct. Mater. 2006, 16, 1608-1614. DOI: 10.1002/adfm.200500855.
  • Kamonwanon, P.; Yodmongkol, S.; Chantarachindawong, R.; Thaweeboon, S.; Thaweeboon, B.; Srikhirin, T. Wear resistance of a modified polymethyl methacrylate artificial tooth compared to five commercially available artificial tooth materials. J. Prosthet. Dent. 2015, 114, 286-292. DOI: 1016/j.prosdent.2015.01.013.
  • Torres, J. M.; Stafford, C. M.; Vogt, B. D. Impact of molecular mass on the elastic modulus of thin polystyrene films. Polymer, 2010, 51, 4211-4217. DOI: 10.1016/j.polymer.2010.07.003.
  • Masood, M. T.; Guerrero, J. A. H.; Ceseracciu, L.; Palazon, F.; Athanassiou, A.; Bayer, I. S. Superhydrophobic high impact polystyrene (HIPS) nanocomposites with wear abrasion resistance. Chem. Eng. J. 2017, 322, 10-21. DOI: 10.1016/j.cej.2017.04. 007.
  • Bhanvase, B. A.; Pinjari, D. V.; Gogate, P. R.; Sonawane, S. H.; Pandit, A. B. Synthesis of exfoliated poly(styrene-co-methyl methacrylate)/montmorillonite nanocomposite using ultrasound assisted in situ emulsion copolymerization. Chem. Eng. J. 2012, 181, 770-778. DOI: 10.1016/j.cej.2011.11.084.
  • Okamoto, M.; Morita, S.; Taguchi, H.; Kim, Y. H.; Kotaka, T.; Tateyama, H. Synthesis and structure of smectic clay/poly(methyl methacrylate) and clay/polystyrene nanocomposites via in situ intercalative polymerization. Polymer 2000, 41, 3887-3890. DOI: 1016/S0032-3861(99)00655-2.
  • Xu, J. S.; Ke, Y. C.; Zhou, Q.; Hu, X. L. In-situ encapsulating MMT intermediate particles by suspension polymerization of poly (methyl methacrylate-co-styrene): preparation, tunable dispersion and properties. J. Polym. Res. 2013, 20, 196. DOI: 10.1007/s10965-013-0196-3.

  • . The proppant size is also a very important property. Especially in deep reservoirs, the width of hydraulic fracture tends to be smaller than that in shallow reservoirs due to higher in-situ stress. Sometimes the small-size proppants are preferred since they can flow into narrower hydraulic fractures. It will be a good addition if the authors can discuss the implications and challenges of introducing polymer for controlling proppant size, especially the small size during the fabrication process.

Response: Thanks for the reviewer’s valuable suggestion, the proppant size is really an important properties for hydraulic fracturing, especially in deep reservoirs. The size of the proppant prepared with suspension polymerization can be facile and easily controlled, which has been discussed in the revised manuscript.

In the revised manuscript:

As we all know, ultra-low density cross-linked polymer beads with very high sphericity can be obtained with suspension polymerization, and this technology has been widely used in the industry, such as the scale production of ion exchange resin [45] and chromatographic column packing [46]. In addition, the size of the proppant prepared with suspension polymerization can be facile controlled by the stirring speed and dispersant concentration [45,46]. This will benefit for the enhancing of conductivity in deep reservoirs, where small-size proppants are preferred since they can flow into narrower fractures (Figure 8) [9].

Figure 8. Hydraulic fracture propped with proppant of different size [9].

  • . What is the difference between the sphericity and roundness of proppants? If these two concepts are identical, the authors should consider using only one word to avoid further confusion. If they are different, the authors should consider explaining the differences.

Response: Thanks for the reviewer’s valuable suggestion, the difference between the sphericity and roundness was explained in the introduction.

In the revised manuscript:

Proppant is a solid particle with certain sphericity and roundness, strength, corrosion and stability, which can prevent the reclosing of fractures generated by hydraulic fracturing, maintain the smooth flow of oil and gas, and improve the productivity [4,5]. In the hydraulic fracturing process, the shape, strength, acid solubility and turbidity can influence the integrity of the newly created fractures, thus determined the efficiency of oil and gas flow out of the well [6]. The sphericity and roundness are very important because it can influence the strength of the proppant. The sphericity is the degree of the proppant close to the sphere, and the roundness refers to the relative sharpness of proppant edges and corners. The angular and pointed proppant tends to break easily, whereas the well-rounded proppant will be mechanically stable. The proppant possesses high sphericity and roundness can provide high conductivity, because larger passages can be formed for this kind of proppant. The ability to withstand compressive loading is also necessary for proppant. The proppant mustn’t break into fines and block the fractures, and slowing down the production rate, thus the proppant should possesses low crushing rate. In addition, the proppant must endure harsh environment such as acid mixture pumped into wells to generated crack, thus the low acid solubility and low turbidity is also very important for the proppant.

  • . There are several typos and grammar errors in this paper. The authors should improve their English writing. The identified typos and errors include but are not limited to:

Response: Thanks for the reviewer’s valuable suggestion, all of other typos and grammar errors in the whole manuscript was revised.

(a)Line 28: prappant -> proppant

The word “prappant” has been revised to “proppant”.

(b)Line 44: carring -> transport

The word “carring” has been revised to “transport”.

(c)Line 53: need -> needed

The word “need” has been revised to “needed”.

(d)Line 55, 84: strengthen -> strength

The word “strengthen” has been revised to “strength”.

(e)Line 64: preperties -> properties

The word “preperties” has been revised to “properties”.

(f)Line 74: "This cause the displacement or viscosity of fracturing fluid needs to be increased" should be rephrased to be grammatically correct.

This sentence has be revised to “To enhance the carrying ability for high density ceramic proppant, the increase of displacement or viscosity for the fracturing fluid is always needed, while this will bring about the problems of difficult treatment of fluid flow back and great damage to oil and gas reservoirs [18]”

(g)Line 91: broken -> breaks

The word “broken” has been revised to “breaks”.

(h)Line 111: rejected -> injected

The word “rejected” has been revised to “injected”.

(i)Line 114: soil -> clay or shale

The word “soil” has been revised to “clay or shale”.

(j)Line 128: decrease -> decreased

The word “decrease” has been revised to “decreased”.

(k)Line 129: reduced -> reduce

The word “reduced” has been revised to “reduce”.

(l)Line 144: test -> tested

The word “test” has been revised to “tested”.

(m)Line 148: obvious improve -> obviously improved

The phrase “obvious improve” has been revised to “obviously improved”.

(n)Line 250: 1.75 -> 1.75%

The word “1.75” has been revised to “1.75%”.

(o)Line 291: obtained -> obtain

The word “obtained” has been revised to “obtain”.

(p)Line 293: spherricity -> sphericity

The word “spherricity” has been revised to “sphericity”.

Reviewer 2 Report

The paper “Progress of Polymer Application in Low-Density Proppant” presents a review  of polymer application to proppant to essentially reduce density. Authors systematically describe various polymers that are used to coat particles as well as provide the resulting properties and references to the original research. Overall, I think that this is a interesting summary that is worth being published.

Comments:

1. Authors are encouraged to provide more motivation on why there is such a significant effort in attempting making lightweight proppant. One reason is related to particle settling in the fractures. The second reason could be the effect of “proppant inertia”, i.e. inability of particles to turn from the wellbore to perforations, see e.g SPE-179117-MS.

2. Another paper that is not directly related, but is worth mentioning is IPTC-19680-MS, where authors describe self-propping mechanism to keep the fractures open instead of using proppant. While I agree that this is not related to polymer coating, but this is an alternative method to solve the problem of high density sand.

3. Typos: prappant on line 9, ceremic on line 308.

Author Response

Response to the Reviewer 2 

The paper “Progress of Polymer Application in Low-Density Proppant” presents a review  of polymer application to proppant to essentially reduce density. Authors systematically describe various polymers that are used to coat particles as well as provide the resulting properties and references to the original research. Overall, I think that this is a interesting summary that is worth being published.

Comments:

  1. Authors are encouraged to provide more motivation on why there is such a significant effort in attempting making lightweight proppant. One reason is related to particle settling in the fractures. The second reason could be the effect of “proppant inertia”, i.e. inability of particles to turn from the wellbore to perforations, see e.g SPE-179117-MS.

Response: Thanks for the reviewer’s valuable suggestion. The introduction was revised according to the reviewer’s suggestion, and relative reference was cited.

In the revised manuscript:

Since the natural sand, glass bead and nutshell has been used as proppants in the hydraulic fracturing, new and efficient proppants have been developed rapidly [7]. After 1970s, due to the excellent performance in oil exploitation, synthetic ceramics proppants sintered with bauxite as the main materials was rapid developed and promoted. In order to further improve the comprehensive properties, polymer coated sand and coated ceramic proppant were developed after 1980s [8]. However, traditional natural sand and synthetic ceramics is not suitable for the eco-friendly hydraulic fracturing, because they are settle quickly in low-viscosity fracturing liquid (Figure 2) [9]. In addition, the inertia of these high density proppants makes them hard to turn from the wellbore to perforations [10]. 

Figure 2. Prop effect of (a) traditional proppant and (b) low-density proppant [9].

The oil and gas production is highly correlated with the propped length and area, and the fracture conductivity of hydraulic fracturing. As reported in the literature [11], 3D fracturing model shows that ULDP cover higher and longer fracture areas with a smaller propped width. The ULDP can improve the propped length and area in low permeability shale reservoirs, while mainly improve the propped area in high permeability shale reservoirs. On the other hand, the fracture conductivity was highly dependent on the proppant size, flow back, and fines generation [6]. Generally, large proppant size, low flow back and low fines generation can provide higher fracture conductivity.

Ref.:

[10] Wu, C. H.; Sharma, M. M. Effect of perforation geometry and orientation on proppant placement in perforation clusters in a horizontal well. SPE Hydraulic Fracturing Technology Conference, Woodlands, Texas, USA. 2016, DOI: 10.2118/179117-MS.

  1. Another paper that is not directly related, but is worth mentioning is IPTC-19680-MS, where authors describe self-propping mechanism to keep the fractures open instead of using proppant. While I agree that this is not related to polymer coating, but this is an alternative method to solve the problem of high density sand.

Response: Thanks for the reviewer’s valuable suggestion. The introduction was revised according to the reviewer’s suggestion, and relative reference was also cited.

In the revised manuscript:

As the traditional hydraulic fracturing fluid possesses a low viscosity, which has a poor carrying ability for high density proppant, such as natural sand, ceramic and glass bead, which has high density and high fines generation [12]. In order to improve the carrying capacity of traditional proppant, it is necessary to increase the displacement or viscosity of fracturing fluid. Though these methods can solve the problem of carrying ability to some extent, yet they will cause new problems such as difficult treatment of fluid flow back and great damage to reservoirs [13]. Therefore, in order to thoroughly solve the problem of proppant transport, it is of great significance to develop novel proppant with low density, or new fracturing technology such as liquid self-propping [14]. According to the difference of density, proppants can be divided into following categories (Table 1) [15].

Ref.:

[14] Pei, Y. X.; Zhou, H. X.; Li, D. P.; Zhang, S. C.; Tian, F. C.; Zhao, L. Q. A novel unconventional reservoir fracturing method-liquid self-propping fracturing technology. International Petroleum Technology Conference, Dhahran, Kingdom of Saudi Arabia. 2020, DOI: 10.2523/IPTC-19680-MS.

  1. Typos: prappant on line 9, ceremic on line 308.

Response: Thanks for the reviewer’s valuable suggestion, “prappant” and “ceremic” have been revised to “proppant” and “ceramic”, respectively. In addition, all of other typos and grammar errors in the whole manuscript was revised.

Round 2

Reviewer 1 Report

Thank authors for addressing my previous comments. My comments and suggestions for the revised manuscript are summarized below.

  1. The authors should consider improving some figures.
    1. In Figures 5, 6, 7, and 9, the figure caption says "Adapted from Gao et al. Unpublished". Are these figures from an internal report of the authors' group? If this is the case, the authors do not need to include this explanation in the figure caption.
    2. For Figures 5 and 7, the authors should add a scale to show the size of the proppant.
    3. For Figures 6 and 9, the authors should improve the scale. There is no unit for the length.
  2. The authors still need to improve their English writing. I suggest that the authors should ask a professional writer or editor to proofread this paper and correct all the syntax errors. An incomplete list of typos or syntax errors is summarized below.
    1. Line 31, should "corrosion" be changed to "corrosion resistance"?
    2. Line 34, "of the proppant" should be added after "turbidity".
    3. Line 40, "possesses" should be changed to "with".
    4. Line 44, "possesses" should be "possess".
    5. Lines 44-47 should be rephrased. The sentence is not syntactically correct.
    6. Line 71, the authors should consider replacing "displacement" with "flow rate" or "pump rate".
    7. Line 214, "enhance the" is repeated twice and should be deleted.

Author Response

Thank authors for addressing my previous comments. My comments and suggestions for the revised manuscript are summarized below.

(1)The authors should consider improving some figures.

â‘ In Figures 5, 6, 7, and 9, the figure caption says "Adapted from Gao et al. Unpublished". Are these figures from an internal report of the authors' group? If this is the case, the authors do not need to include this explanation in the figure caption.

Response: Thanks for the reviewer’s valuable suggestions. These figures are from our internal report, and the explanation in the figure captions was deleted in the revised manuscript.

â‘¡For Figures 5 and 7, the authors should add a scale to show the size of the proppant.

Response: Thanks for the reviewer’s valuable suggestions. The scale bar was added in Figure 5 and 7 to show the sized of the proppant.

In the revised manuscript:

Figure 5. Sand (a) and coated sand (b) proppant.

Figure 7. Nutshell (a) and coated nutshell (b) proppant.

â‘¢For Figures 6 and 9, the authors should improve the scale. There is no unit for the length.

Response: Thanks for the reviewer’s valuable suggestions. The unit for the scale bar was added in Figure 6 and 9.

In the revised manuscript:

Figure 6. Ceramic (a) and coated ceramic (b) proppant.

Figure 9. ULDP based on polymer composite beads.

  • The authors still need to improve their English writing. I suggest that the authors should ask a professional writer or editor to proofread this paper and correct all the syntax errors. An incomplete list of typos or syntax errors is summarized below.

Response: Thanks for the reviewer’s valuable suggestion. The grammar and typos in the whole manuscript was revised by a colleague with overseas study experience, and the whole manuscript was revised to our best.

â‘ Line 31, should "corrosion" be changed to "corrosion resistance"?

The word “corrosion” has been revised to “corrosion resistance”.

â‘¡Line 34, "of the proppant" should be added after "turbidity".

The phrase “of the proppant” has been added after “turbidity”.

â‘¢Line 40, "possesses" should be changed to "with".

The word “possesses” has been revised to “with”.

â‘£Line 44, "possesses" should be "possess".

The word “possesses” has been revised to “with”.

⑤Lines 44-47 should be rephrased. The sentence is not syntactically correct.

The sentence has been revised to “The proppant mustn’t break into fines and block the fractures, which will down the production rate, thus the proppant should with low crushing rate.”

â‘¥Line 71, the authors should consider replacing "displacement" with "flow rate" or "pump rate".

The word “displacement” has been revised to “flow rate”.

⑦Line 214, "enhance the" is repeated twice and should be deleted.

The repeated phrase “enhance the” was deleted.
